# Transcriptome sequencing and screening of genes related to the MADS-box gene family in *Clematis courtoisii*

**Mingjian Chen** [ORCID], **Linfang Li** *, **Shu'an Wang, Peng Wang, Ya Li** [ORCID] *

Department of Ornamental Plant Research Center, Institute of Botany, Jiangsu Province and Chinese Academy of Sciences, Nanjing, China

* lilinfangqq@cnbg.net (LL); yalicnbg@cnbg.net (YL)

**Data Availability Statement:** All data generated or analyzed during this study are within the paper and its Supporting Information files.

## Abstract

The MADS-box gene family controls plant flowering and floral organ development; therefore, it is particularly important in ornamental plants. To investigate the genes associated with the MADS-box family in *Clematis courtoisii*, we performed full-length transcriptome sequencing on *C. courtoisii* using the PacBio Sequel third-generation sequencing platform, as no reference genome data was available. A total of 12.38 Gb of data, containing 9,476,585 subreads and 50,439 Unigenes were obtained. According to functional annotation, a total of 37,923 Unigenes (75.18% of the total) were assigned with functional annotations, and 50 Unigenes were identified as MADS-box related genes. Subsequently, we employed hmmerscan to perform protein sequence similarity search for the translated Unigene sequences and successfully identified 19 Unigenes associated with the MADS-box gene family, including MIKC*(1) and MIKC$^C$ (18) genes. Furthermore, within the MIKC$^C$ group, six subclasses can be further distinguished.

## Introduction

The genus *Clematis* (Ranunculaceae) contains mostly woody vines, a few perennial herbs, subshrubs and shrubs [1]. *C. courtoisii* is a deciduous woody vine with opposite, ternate, or pinnate leaves and large, white flowers that bloom in spring and summer from May to June. In addition, it is a vine with ornamental value, and it is an excellent parent for innovative ornamental Clematis varieties [2, 3]. There are relatively few studies on the key genes of flowering regulatory mechanisms, which limit flowering processes and new variety breeding.

The MADS-box gene family is a large transcription factor (TF) family, and the members play important roles in the regulation of plant flowering and flower organ growth and development [4, 5]. An increased understanding of flowering gene-related information is conducive to improving ornamental plant flower traits. Based on phylogenetic relationships, plant MADS-box genes are classified into type I and type II. The plant type I MADS-box genes each encode a protein containing a highly conserved SRF-like MADS domain, whereas the plant type II MADS-box genes each encode a protein containing a MEF2-like MADS domain. Additionally, the type II MADS-box domain is more conserved than the type I MADS-box domain

**Funding:** Funding This work was supported by the National Natural Science Foundation of China (31800603), Jiangsu Key Laboratory for the Research and Utilization of Plant Resources (JSPKLB202203), and the Technology Innovation Alliance of Flower Industry Fund(2020hhlm004).

**Competing interests:** The authors have declared that no competing interests exist.

[6]. In addition to the MADS-box type II domain, these proteins also contain K-box, Intervening, and C-terminal domains [7], whereas the MADS-box type I proteins lacks a K-box domain. Different gene structures and translated protein structures allow type I to be divided into three subgroups: Mα, Mβ, and Mγ, whereas type II can be divided into two subgroups: MIKC$^C$ and MIKC*. MIKC$^C$ genes in angiosperms mainly belong to 13 branches, which can be divided into 12 major subgroups in *Arabidopsis thaliana*: AG, AGL6, AGL12, AP3/PI, GGM13, STMADS11, TM3, AGL2/SEP, AGL17, AP1/SQUA, AGL15, and FLC [8, 9], and they participate in the current ABCDE floral organ model.

Transcriptome sequencing provides a wealth of information on RNA transcripts, and its high accuracy has allowed its wide application. Sequencing technology has been used in the study of Clematis genetic resources [10], transcriptome [11–13], flower type, flower color formation mechanism [14], and molecular markers [15]. However, owing to the lack of a Clematis genome and transcriptome sequence information, sequencing research lags behind that of other species.

The third-generation sequencing technology, represented by PacBio, uses Single Molecule Real-Time (SMRT) sequencing to overcome the limitations of second-generation sequencing, such as reliance on template amplification and sequence read length limitation, with the advantage of ultra-long sequencing read length, it can cover high repeat and low complexity regions to detect repetitive genomes and structural variation regions, directly read the full-length cDNA of reverse transcription, obtain full-length transcript sequences, and accurately and completely reflect the sequence information of species [16]. Complete transcripts can detect variable splice isoforms, lead to the discovery of more splice sites to complement genome annotation, and can be used to better analyze transcript structures [17, 18]. The advantages of SMRT sequencing technology can be used to aid in the study of transcriptome information for non-model plants lacking a reference genome, including soybean [19], wheat [20], and cabbage [21].

Therefore, PacBio SMRT technology was employed in this study to acquire full-length transcriptome information of *C. courtoisii*. By comparing with the database, functional annotation information of Unigenes was obtained, and through domain search of translated protein sequences, 19 MADS-box gene family related Unigenes were identified and the conserved domain composition of translation proteins was analyzed. A phylogenetic tree was constructed using homologous protein sequence alignment and phylogenetic analysis. The results provide data for the subsequent study of biological functions, molecular mechanisms, and related functional Clematis genes.

## Materials and methods

The experimental material was grown in the Clematis nursery of the NANJING BOTANICAL GARDEN MEM. SUN YAT-SEN (Nanjing, China) and was identified as 3-year-old *Clematis courtoisii* Hand. -Mazz (Fig 1) by Dr. Wang Shu' an. The voucher specimen was deposited in the herbarium at the Institute of Botany, Jiangsu Province and Chinese Academy of Science. The certificate number is No. 0626817. The tissues of root, stem, leaf, and flower were each sampled three times and stored at -80°C for subsequent experiments.

### Total RNA extraction and SMRT library construction

Total RNA was extracted and purified using RNApure kit (HUAYUEYANG, China). The purity of total RNA was detected by OneDrop2000 (WINS, China). The RNA concentrations were accurately quantified using Qubit4.0 (Invitrogen, USA). A 2100 bioanalyzer (Agilent, USA) was used to accurately determine RNA integrity. The mRNA was reverse-transcribed

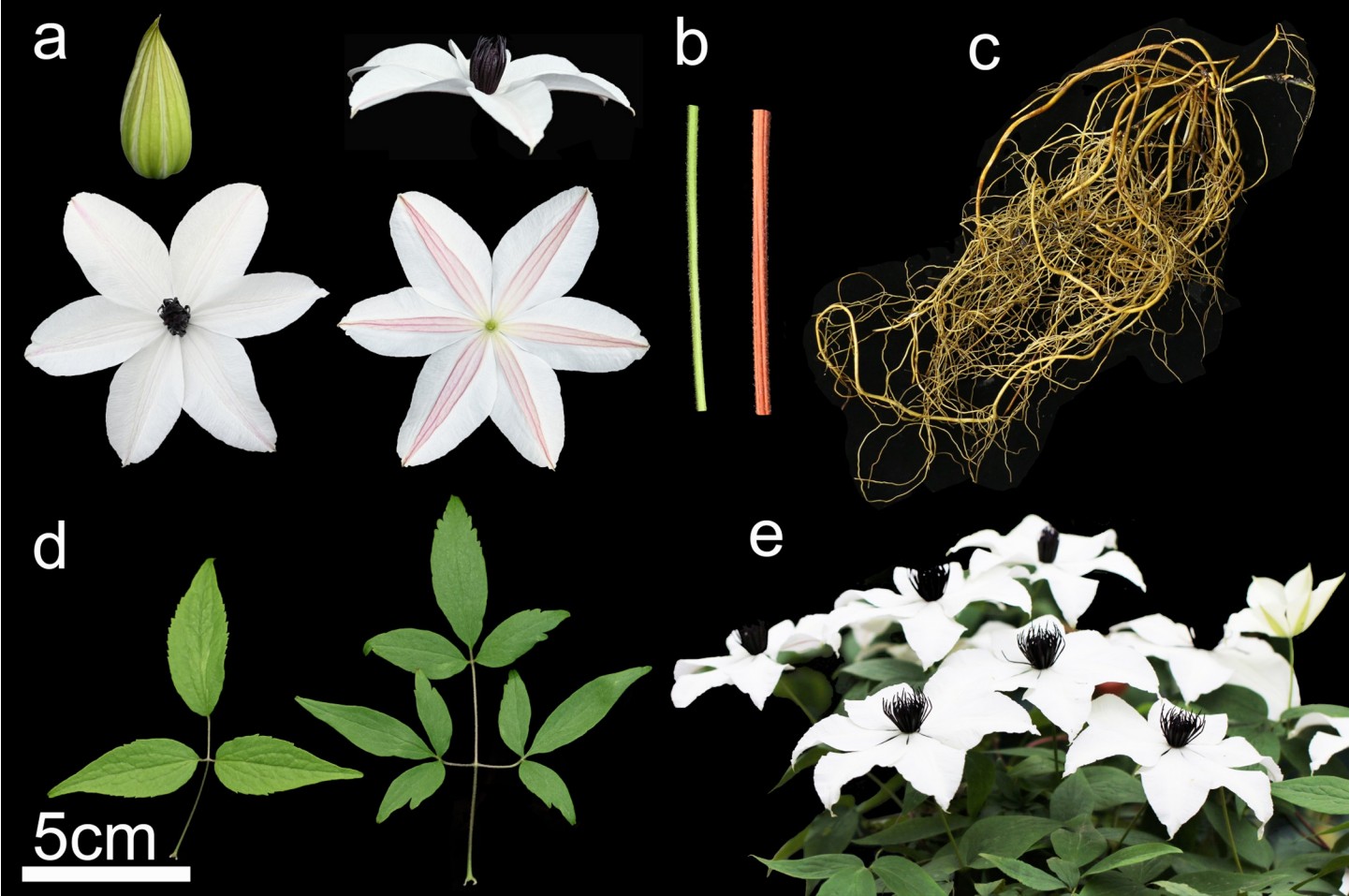

**Fig 1. Introduced *C. courtoisii* in the Clematis nursery.** (**a**) flowers; (**b**) stem; (**c**) roots; (**d**) leaves; (**e**) appearance. Photographed by Dr. Wang Shu' an.

into cDNA using the HiScript III 1st Strand cDNA Synthesis Kit (Vazyme, China). Subsequently, the amplified and enriched cDNA underwent PCR, followed by damage repair and terminal repair of the full-length cDNA. Next, SMRT junction and exonuclease digestion were performed for connection, and quality detection was conducted using Qubit4.0 and A2100. SMRT library sequencing of *C. courtoisii* was completed by Genepioneer Biotechnologies (Nanjing, China).

All the raw data has been uploaded to the National Center for Biotechnology Information Sequence Read Archive (www.ncbi.nlm.nih.gov) under accession number SUB13428639.

### Transcriptome data processing and error correction

The raw data was analyzed using SMRT Linkv5.0.0 to obtain Subreads. The circular consensus sequence (CCS) was obtained by correcting the single-molecule multiple sequencing reads of Subreads, and then, the full-length non-chimera sequence (flncs) were identified according to whether the CCS contained cDNA primers and poly (A) tails. Non-chimeric ROI sequences were divided into full-length ROIs (the presence of poly (A) tail signals, 5′ adaptors, and 3′ adapter sequences) or non-full-length ROIs. The full-length sequences were clustered without redundancy and corrected using non-full-length sequences. Consensus reads were obtained

after clustering, which will serve as reference transcript sequences for subsequent analyses. The full-length transcripts were clustered using Blat [22], and the similar transcripts meeting the criteria of e-value = 1e-10, min-identity = 90%, and min-coverage = 85% were grouped into a Unigenes.

## Gene functional annotation

We employed the BLAST to perform comparative analysis between Unigenes and a diverse array of publicly available databases, thereby acquiring comprehensive functional annotation information for the Unigenes with default parameters (e-value$<10^{-5}$). The utilized databases encompassed the Non-Redundant Protein Database (NR), Gene Ontology (GO), Clusters of Orthologous Groups (COG), euKaryotic Orthologous Groups (KOG), Swiss-Prot protein database, Kyoto Encyclopedia of Genes and Genomes (KEGG), as well as Pfam.

## Identification of MADS-box genes

We initially identified MADS-box related genes based on the annotation information of Unigenes. Subsequently, we employed the SMART database (http://smart.embl-heidelberg.de/) to detect the conserved domain of MADS in the translated protein sequences of these genes. The domain information for MADS-box gene SRF-TF (PF00319) and K-box (PF01486) was obtained from the Pfam database, and Unigenes translation protein sequences were searched using HMMER (V3.0)'s hummscan function [23] with default parameters (e-value$<0.01$).

## Phylogenetic tree construction of MADS-box transcription factor genes

The candidate MADS-BOX gene translation proteins of *C. courtoisii* were compared with the NCBI database, and a selection of 45 MADS proteins from model plants Arabidopsis thaliana, Paeonia suffruticos, Nelumbo nucifer, and Vitis vinifera was made based on high confidence levels. We performed multiple sequence alignment for the MADS-box gene translation protein using Clustal W2.1 software with default parameter settings. Phylogenetic trees were constructed using the MEGA 11.0 maximum likelihood algorithm with a Bootstrap value of 1000 and default values for other parameters.

## Conserved domain analysis of MADS-box transcription factor family member

The conserved domain of the candidate MADS-box gene translation protein was verified using Batch CD-Search (https://www.ncbi.nlm.nih.gov/Structure/bwrpsb/bwrpsb.cgi). Subsequently, we analyzed the conserved motif for the candidate MADS-box gene translation protein using MEME (http://meme.nbcr.net/meme/Intro.html) with the following parameters: a maximum of 6 discoveries and a maximum motif length of 75.

## Results and analysis

### Statistical analysis of PacBio Iso-Seq data

The full-length transcriptome database of *C. courtoisii* contains 12.38 Gb of Subreads bases and 9,476,585 subreads, with the longest and shortest sequences being 81,141 bp and 50 bp, respectively (Table 1). After the IsoSeq calibration process, 422,635 CCSs, 327,248 full-length reads, 95,359 non-full-length reads, 300,171 flncs, and 120,294 consensus sequences were obtained. After filtering out potential false alignments and redundant transcripts, 120,293 non-redundant transcripts were acquired for the following analysis. The total length of the transcript was 162,573,047 bp, with the average length being 1,178 bp, the N50 length being

**Table 1. Summary of PacBio IsoSeq data from *C. courtoisii*.**

| samples | T01 |
| --- | --- |
| Subreads base(G) | 12.38 |
| Subreads number | 9,476,585 |
| Average length | 1,306 |
| N50 | 1,455 |
| CCS-reads | 422,635 |
| nfl-reads | 95,359 |
| fl-reads | 327,248 |
| flnc-reads | 300,171 |
| Mean-flnc | 1,379 |
| consensus | 120,294 |

1,605 bp, and the GC content being 44.64%. After de-redundancy, 50,439 Unigenes were obtained. The total length of the Unigenes was 78,967,817 bp, the average length was 1,390 bp, the N50 length was 1,869 bp, and the GC content was 44.36%. The length distributions of Subreads, Flncs, Consensus, and Unigenes are shown in Fig 2.

## Functional annotation

The functional annotation of the Unigenes was conducted on the basis of sequence similarities in seven databases. Among the 50,439 Unigenes, 37,923 (75.18%) Unigenes were annotated. In total, 1,936 Unigenes were consistently annotated in the seven databases, and 12,516 Unigenes were not annotated (Table 2).

The annotation results for the Unigenes in the NR database revealed that *Nelumbo nucifera* 7,154 (19.06%), follow by *Vitis vinifera* 3,136 (8.36%), had the largest number of matched genes to the Unigenes of *C. courtoisii*. The number of matching genes from other species did not exceed 1,000 (Fig 3), and the number of matched genes in *Clematis* L. is even smaller.

The KOG database annotation results showed that many Unigenes were involved in most life activities (Fig 4). The largest number of genes were involved in general function prediction (3,996, 18.50%), follow by post-translational modification, protein inversion and chaperone genes (2,996, 13.87%), signal transduction mechanism genes (1,834, 8.49%), translation, ribosomal structure and biogenesis genes (1,697, 7.86%), carbohydrate transport and metabolism (1,496, 6.93%), amino acid transport and metabolism (1,257, 5.82%), and energy generation and conversion (1,251, 5.79%). RNA processing and modification (1,053, 4.88%), intracellular transport, secretion and vesicle transport (1,052, 4.87%), and transcription (1,041, 4.82%). In addition, 1,088 (5.04%) genes were annotated to unknown functions.

The GO annotation analysis (Fig 5) revealed the following: The number of genes related to cellular component was 32,061, which belonged to 15 lower classifications; The number of genes related to biological process was 43171, which belonged to 19 lower classifications; The number of genes related to molecular function was 14,944, which belonged to 17 lower classifications. The number of Unigenes involved in cellular and metabolic processes were the highest, at 8,843 and 10,920, respectively. The single biological process class had 7,252 Unigenes. Among the cell components, the Unigenes of cell and cell parts were the most, at 8,019 and 8,240, respectively. The numbers of membrane and organelle were 3,234 and 6,011, respectively. Among the molecular functions, the catalytic activity Unigenes were the most, at 7,767. The combination class was 5,491.

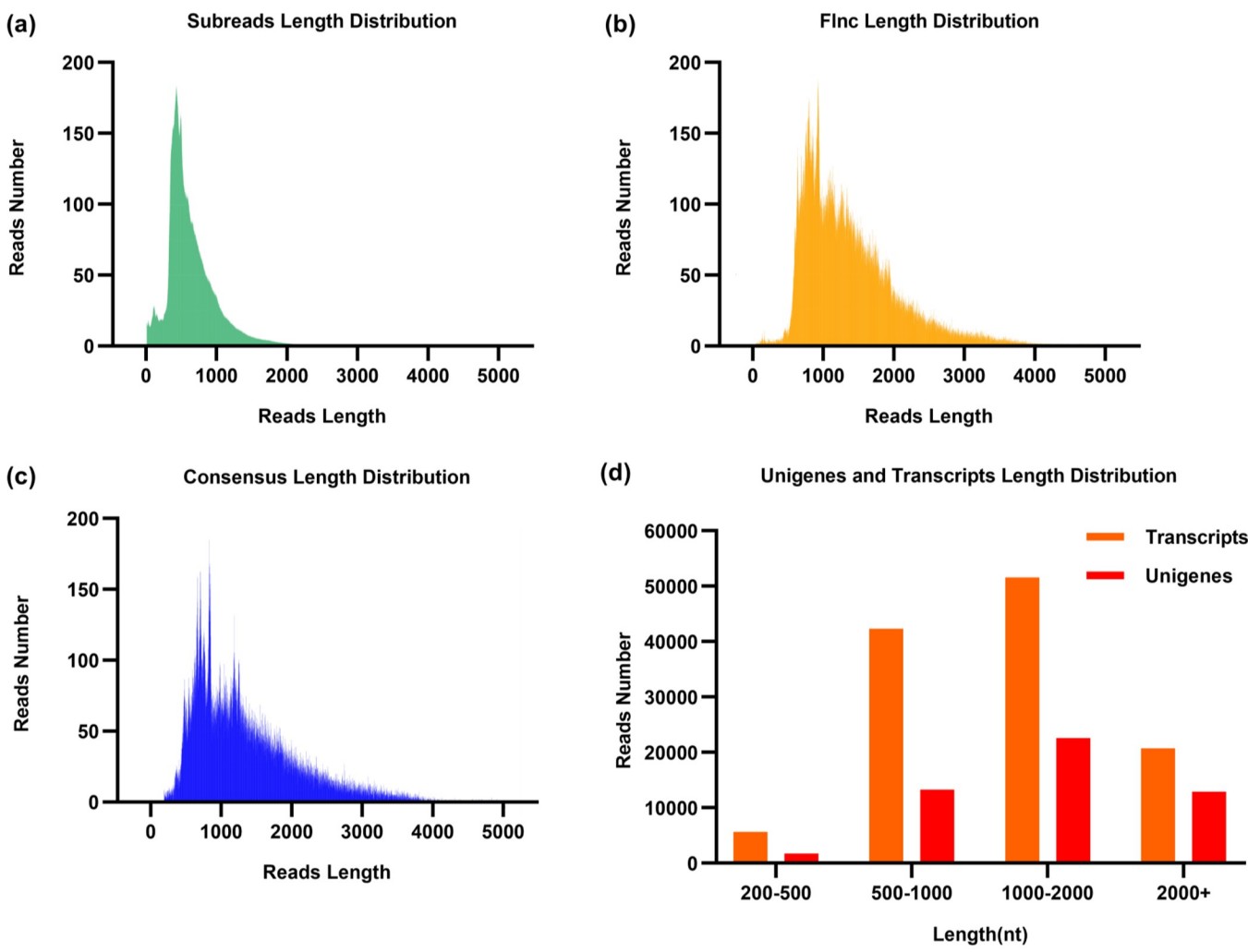

**Fig 2. From the full-length transcriptome PacBio Iso-Seq data of *C. courtoisii*.** (**a**) Subread; (**b**) Flnc; (**c**) Consensus; and (**d**) Unigene and Transcript length distributions.

**Table 2. Summary of the functional annotation.**

| Annotated Database | Annotated Number | Percentage (%) |
|---|---|---|
| COG Annotation | 9,755 | 19.34 |
| GO Annotation | 17,400 | 34.50 |
| KEGG_Annotation | 12,438 | 24.66 |
| KOG_Annotation | 21,596 | 42.82 |
| Pfam_Annotation | 12,316 | 24.42 |
| Swissprot_Annotation | 27,147 | 53.82 |
| nr_Annotation | 37,528 | 74.40 |
| All Annotated | 37,923 | 75.19 |

# Nr Homologous Species Distribution

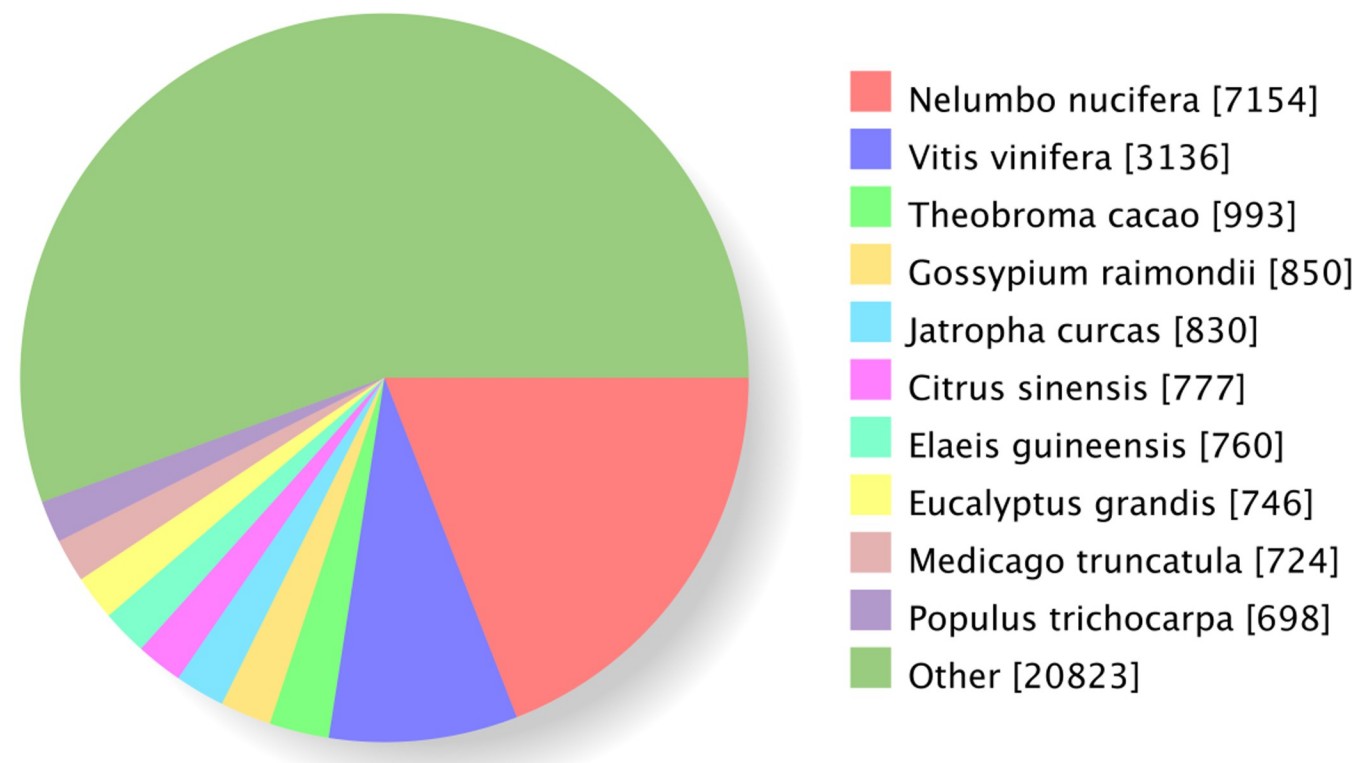

**Fig 3. Homologous species distribution in the non-redundant (Nr) database.**

## The screening of the MADS gene family

We compared the Unigene sequences against Swiss-Prot, NR, Pfam, and KEGG databases to identify 50 Unigenes associated with MADS-box transcription factors. Subsequently, utilizing the SMART database, we further identified the conserved MADS-BOX domain in the translated protein sequences of these Unigenes, resulting in the identification of 19 candidate MADS genes. By employing HMMER to search for the presence of PF00319 and PF01486 domains in the translated protein sequences of Unigenes, we successfully identified 19 candidate MADS genes, as presented in Table 3.

## The classification and construction of phylogenetic trees for the MADS-box gene family

We conducted a comparative analysis of MADS-box protein sequences from *Arabidopsis thaliana* (10), *Paeonia suffruticosa* (9), *Nelumbo nucifera* (11), and *Vitis vinifera* (15) that exhibited significant homology with CcMADS of *C. courtoisii*. Subsequently, we constructed a phylogenetic tree to visualize the relationships between these sequences (Fig 6). Following the classification system for Arabidopsis MADS-box gene family, we identified 19 CcMADS genes as type II MIKC (18) and MIKC* (1). Further investigation revealed that the 18 MIKC genes could be categorized into subfamilies including STMADS11 (4), DEF (1), GLO (2), AGL6 (1),

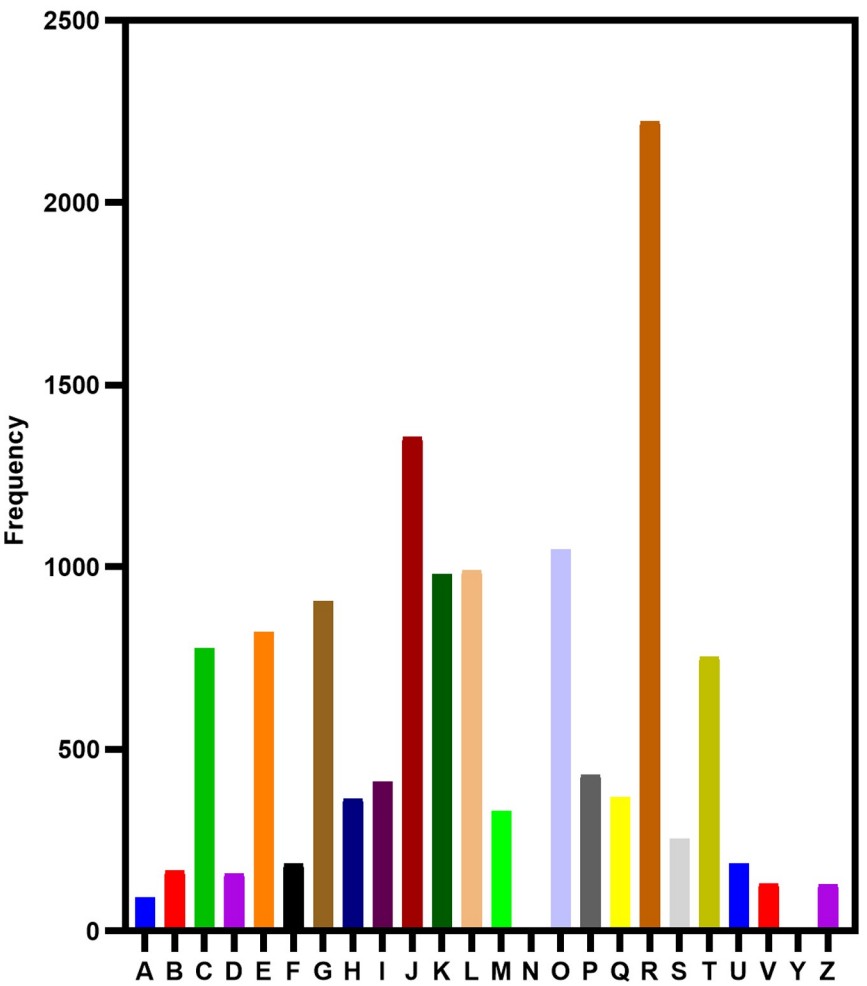

**Fig 4. euKaryotic Orthologous Groups (KOG) classification.**

SQUA (2), and TM3 (8). Among them, CcMADS08 and CcMADS15 exhibited homology to the AP1 gene of the SQUA subfamily, which is associated with flower development class A genes. CcMADS11 and CcMADS14 displayed homology to the PI gene of the GLO subfamily within class B genes, while CcMADS04 showed homology to the AP3 gene of the DEF subfamily.

## The distribution of MADS gene motifs

The structural integrity of MIKC-type MADS domain proteins is crucial for their functional efficacy, with the MADS and K domains playing indispensable roles in DNA binding and protein complex formation, respectively. The genetic protein sequences of the mads-box are analyzed and identified using online meme software. Six relatively conservative basic orders are obtained (Fig 7). The MADS-box domain comprises Motifs 1, 3, 5, and 6, while the K-box domain consists of Motifs 2 and 4. CcMADS05, 07, and 17 exhibit a relatively complete MADS-domain with high conservation levels but a less conserved K-domain; whereas other type II CcMADS proteins solely possess the MADS-domain (Fig 8).

**Cellular Component**

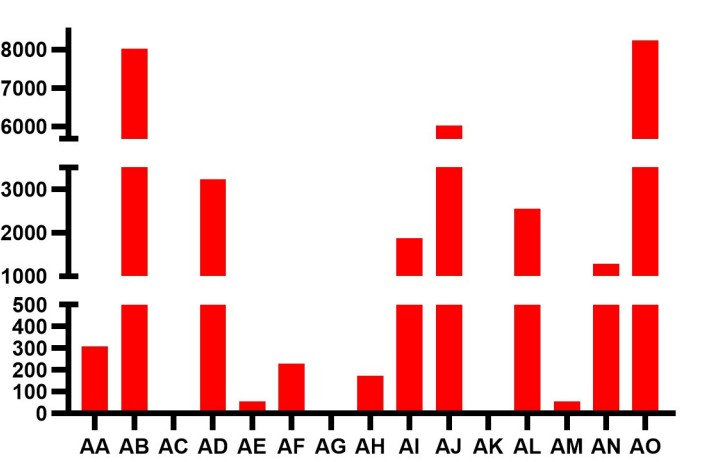

AA:extracellular region
AB:cell
AC:nucleoid
AD:membrane
AE:virion
AF:cell junction
AG:extracellular matrix
AH:membrane-enclosed lumen
AI:macromolecular complex
AJ:organelle
AK:extracellular region part
AL:organelle part
AM:virion part
AN:membrane part
AO:cell part

**Molecular Function**

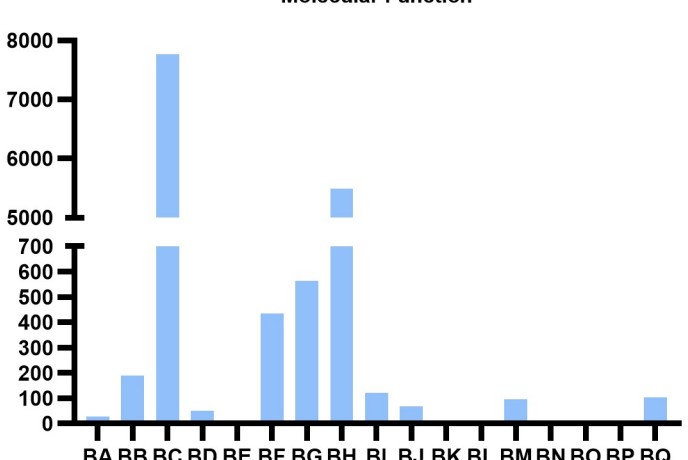

BA:protein binding transcription factor activity
BB:nucleic acid binding transcription factor activity
BC:catalytic activity
BD:receptor activity
BE:guanyl-nucleotide exchange factor activity
BF:structural molecule activity
BG:transporter activity
BH:binding
BI:electron carrier activity
BJ:antioxidant activity
BK:channel regulator activity
BL:metallochaperone activity
BM:enzyme regulator activity
BN:protein tag
BO:translation regulator activity
BP:nutrient reservoir activity
BQ:molecular transducer activity

**Biological Process**

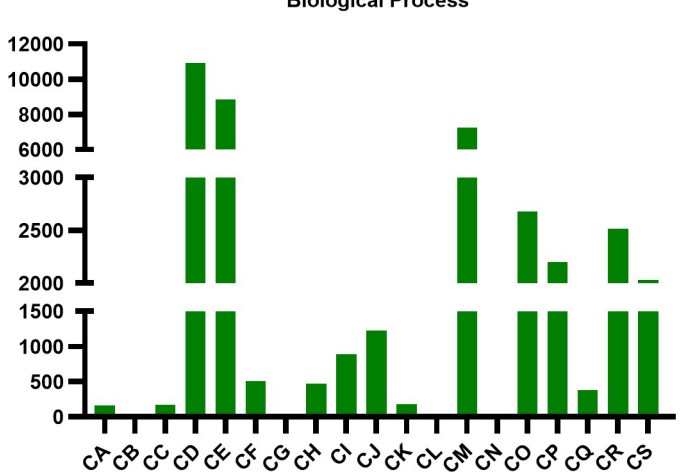

CA:reproduction
CB:cell killing
CC:immune system process
CD:metabio;ic process
CE:cellular process
CF:reproductive process
CG:biological adhesion
CH:signaling
CI:multicellular organismal process
CJ:developmental process
CK:growth
CL:locomotion
CM:single-organism process
CN:rhythmic process
CO:response to stimulus
CP:localization
CQ:multi-organism process
CR:biological regulation
CS:cellular component organization or biogenesis

**Fig 5. Gene Ontology (GO) classification.**

**Table 3. The information of 19 CcMADS genes in *C. courtoisii*.**

| Gene number | Rename | Homologous gene | MADS-box Subclade |
|---|---|---|---|
| **Unigene003539** | CcMADS01 | At1g18750 | MIKC* |
| **Unigene003548** | CcMADS02 | At2g22540 | STMADS11 |
| **Unigene003606** | CcMADS03 | At2g22540 | STMADS11 |
| **Unigene003651** | CcMADS04 | At3g54340 | DEF |
| **Unigene008563** | CcMADS05 | At2g22540 | STNADS11 |
| **Unigene008579** | CcMADS06 | At2g45660 | TM3 |
| **Unigene013455** | CcMADS07 | At2g45650 | AGL6 |
| **Unigene025682** | CcMADS08 | At5g60910 | SQUA |
| **Unigene026642** | CcMADS09 | At2g22540 | STMADS11 |
| **Unigene028043** | CcMADS10 | At2g45660 | TM3 |
| **Unigene029881** | CcMADS11 | At5g20240 | GLO |
| **Unigene031824** | CcMADS12 | At2g45660 | TM3 |
| **Unigene034958** | CcMADS13 | At5g20240 | TM3 |
| **Unigene038940** | CcMADS14 | At1g69120 | GLO |
| **Unigene043979** | CcMADS15 | At2g45660 | SQUA |
| **Unigene046039** | CcMADS16 | At2g45660 | TM3 |
| **Unigene047152** | CcMADS17 | At2g45660 | TM3 |
| **Unigene047373** | CcMADS18 | At1g18750 | TM3 |
| **Unigene049456** | CcMADS19 | At2g22540 | TM3 |

## Discussion

In recent years, with the rapid development of plant genomics and transcriptomics, more plant genetic resources have been utilized and protected [24, 25].

The lack of reference genome information largely limits the in-depth study of a species. With the development of sequencing technology, species genome annotation information has been further supplemented and improved. In this study, the PacBio third-generation sequencing platform was used to sequence the transcriptome of *C. courtoisii* tissue samples, and Unigenes were constructed. Subsequently, the functional annotation and gene structure of the transcriptome, which greatly enriched the transcriptome information for *C. courtoisii*, were analyzed. A total of 50,439 Unigenes was obtained, having an average length of 1,390 bp. There were 35,459 Unigenes having lengths greater than 1,000 bp, which accounted for 70.3% of the total number of sequences. The number and proportion of Unigenes were higher than those of previous studies on the second-generation transcriptome sequencing of *Clematis finetiana* [15], *Clematis apiifolia* [26], *Clematis lanuginosa* [27], and *Clematis crassifolia* [28]. The results show that the sequencing quality is higher and the assembly effect is better.

Gene functional annotation information for *C. courtoisii*. In this study, the functional annotations of *C. courtoisii* Unigenes were obtained by comparisons with multiple public databases. A total of 37,923 Unigenes was annotated, accounting for 75.18% of the total number of sequences, which was significantly higher than those of *C. finetiana*, *C. apiifolia*, *C. lanuginosa* and *C. crassifolia*, indicating that the number and quality of the transcripts obtained by third-generation transcriptome sequencing were higher than those of second-generation sequencing. A total of 37,528 Unigenes was annotated in the NR database, and they were highly similar to the annotation results of the second-generation sequencing transcripts for the above-mentioned Clematis plants. This may be because there are few species of Ranunculaceae Clematis in the database, and the lack of research has led to the transcripts of *C. courtoisii* not being compared to members of the same family. In addition, 12,516 (24.82%) Unigenes were not

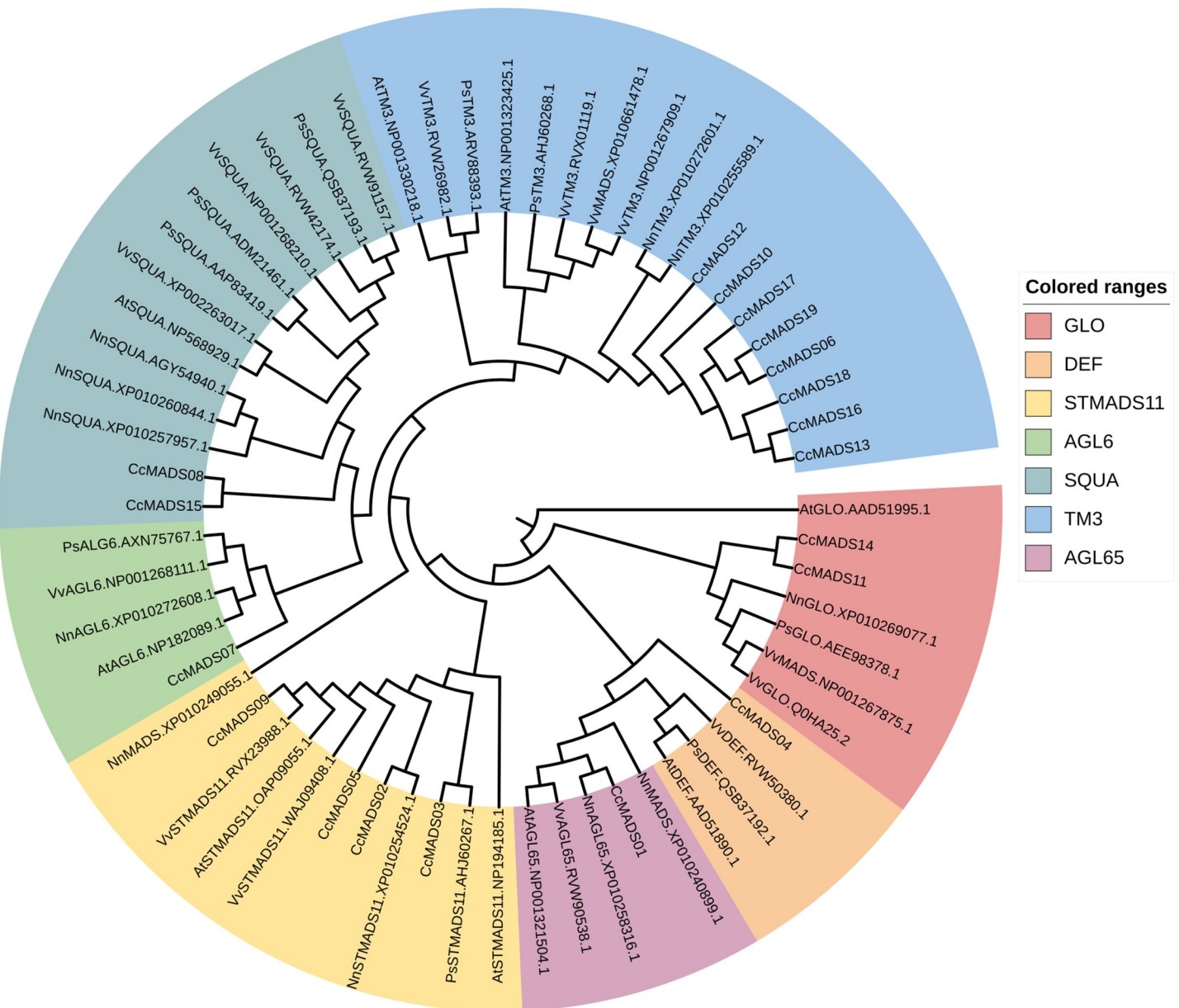

**Fig 6. Phylogenetic tree of MADS-box proteins in *C. courtoisii*, *A. thaliana*, *P. suffruticosa*, *N. nucifera* and *V. vinifera*.** Maximum Likelihood (ML) was used, and the Bootstrap parameter was set to 1,000.

annotated in any database, which may be due to the lack of matched genome and transcriptome information. The overall sequencing results showed that the information obtained by the third-generation sequencing was better than that of the second-generation sequencing.

Based on the functional classification revealed by the mutant phenotype of Arabidopsis MIKC$^C$ gene, Class A genes including AP1, CAL, FUL and AGL79 not only play diverse roles in the regulatory network of flower development but also participate in the formation of floral organs [29]. The B genes mainly consist of AP3 and PI which control petal and stamen

| | Logo | E-value | Sites | Width |
|---|---|---|---|---|
| 1 | | 9.3e-1665 | 43 | 53 |
| 2 | | 5.7e-659 | 33 | 57 |
| 3 | | 1.4e-159 | 35 | 15 |
| 4 | | 3.6e-122 | 22 | 29 |
| 5 | | 7.0e-092 | 16 | 15 |
| 6 | | 3.0e-085 | 11 | 21 |

**Fig 7. The conservation domain of 19 CcMADS genes in *C. courtoisii*.** After conducting gene sequence analysis, we identified six highly conserved motifs. Notably, the gene sequence of motif1 encompasses the gene sequences of both motif5 and 6.

formation [30, 31]. C/D genes such as AG, AGL5 and AGL11 primarily promote stamen and carpel development while participating in ovule formation and inhibiting AP1 gene expression [29]. SEP1, 2, 3 and 4 genes belonging to class E are involved in organogenesis during each round of flower development working together with other MADS-box protein complexes to regulate flower meristem activity as well as organ specificity [32].

In this study, we identified 19 MADS-Box genes of *C. courtoisii*. Based on the phylogenetic tree of MADS homologous proteins in model plants, we found type II MIKC* type (1) and MIKC$^C$ type (18) genes, while no MADS-box gene of type I was detected. The MIKC$^C$ group can be further classified into TM3 (8), STMADS11 (4), DEF (1), GLO (2), AGL6 (1), and SQUA (2) homologous genes. Among them, CcMADS08 and CcMADS15 exhibited homology to the AP1 gene of the SQUA subfamily, which is associated with flower development class A genes. CcMADS11 and CcMADS14 displayed homology to the PI gene of the GLO subfamily within class B genes, while CcMADS04 showed homology to the AP3 gene of the DEF subfamily. The STMADS11 gene family is implicated in the determination of flowering time, seed development, and floral meristem specification [33]. Therefore, the presence of members from this gene family in the transcriptome of *C. courtoisii* may indicate their involvement in regulating these developmental processes. Similarly, the TM3 gene family is involved in organ morphogenesis, stress and hormone signal responses, as well as controlling flowering time [34]. Thus, the presence of TM3 gene members in the transcriptome may suggest their role in regulating these processes. Finally, the AGL6 gene family participates in organ morphogenesis, stress and hormone signal responses and controls flowering time [35]. its presence in the transcriptome implies its regulation over various developmental processes. The identification of these gene members implies that *C. courtoisii* possesses a regulatory mechanism during the process of floral organ development and reproduction.

Long-read third-generation transcriptome sequencing technology enables the generation of full-length transcripts, providing comprehensive coverage of entire gene families [36]. Through alignment and analysis of long-read sequencing data, it is possible to identify gene families with similar sequences. The alignment of conserved protein motifs revealed variations in the number and position of these motifs among homologous reference protein sequences from different subfamilies. Notably, STMADS11 homolog CcMADS05 and CcMADS17, as well as squa homolog CcMADS07, exhibited relatively complete MADS-domain and K-domain structures. However, other CcMADS proteins displayed incomplete MIKC domains

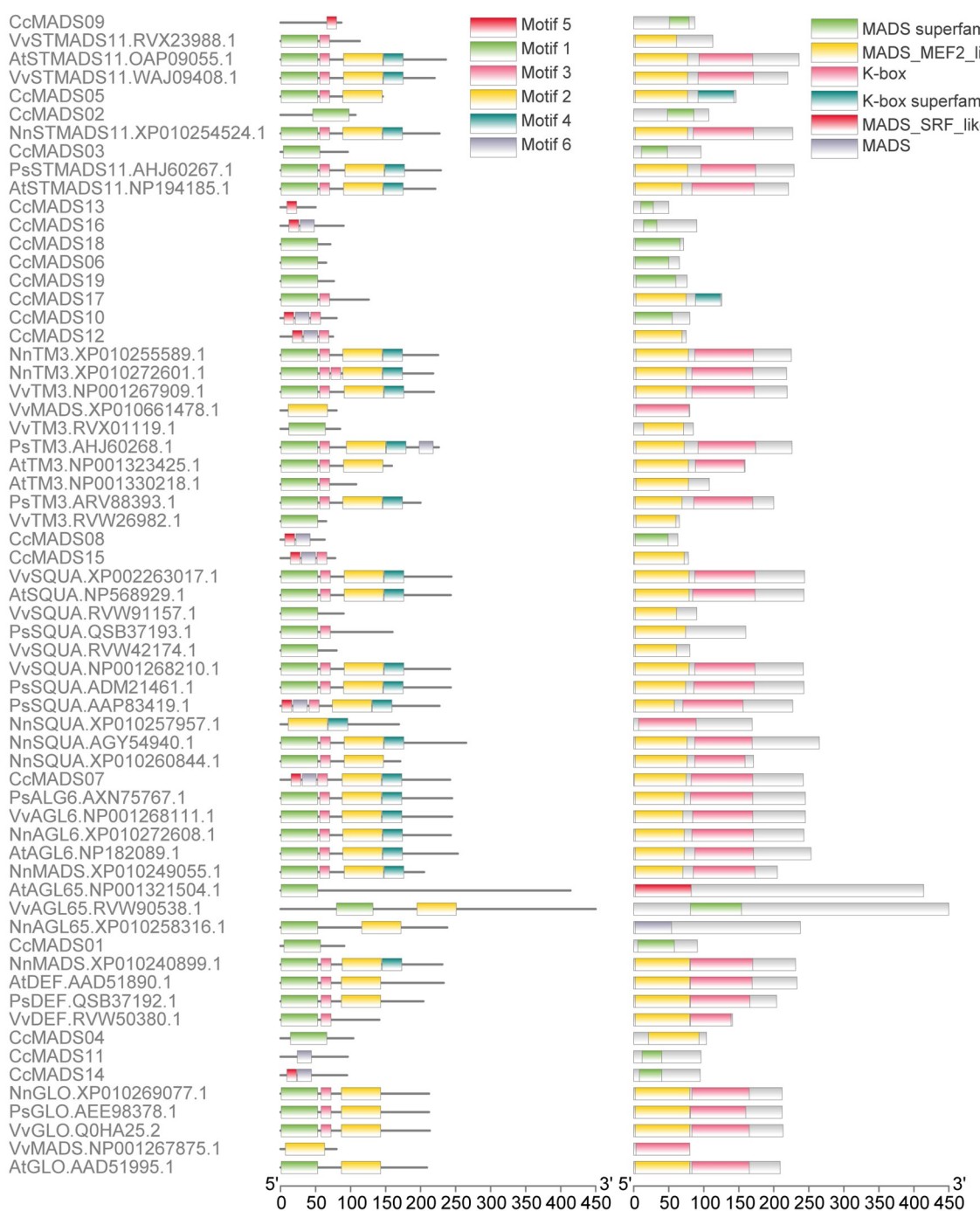

**Fig 8. Conserved sequences of MADS-box proteins in *C. courtoisii*, *A. thaliana*, *P. suffruticosa*, *N. nucifera* and *V. vinifera*.** a maximum of 6 discoveries and a maximum motif length of 75.

due to limitations in existing protein structure databases. Consequently, if the target protein sequence bears low similarity to those present in these databases [37], predicted structures may be incomplete or inaccurate. Additionally, complex genomic structures and repetitive sequences can hinder accurate discrimination of transcriptome sequencing reads [38], which

may lead to the splitting of the same region into multiple incomplete sequences in the cluster deredundancy process. While genome sequencing encompasses coding and noncoding regions of all genes, third-generation transcriptome sequencing specifically captures expressed transcripts at a given time point. Therefore, genome sequencing offers higher coverage and depth compared to transcriptome sequencing for precise identification of gene families within a genome.

Clematis varieties have wide petals, rich colors, stable flowering, and high ornamental value. MADS-box family genes play important roles in regulating the development and differentiation of floral organs in ornamental plants. We can regulate the expression of this family of gene using molecular biology, and the study of regulatory molecular mechanisms of flower type, flowering period, and other traits, which will be of great significance to the popularization and application of Clematis flowers in gardens. In this study, we identified 19 members of the MADS-BOX gene family from the protein sequence data translated from the full-length transcript of *C. courtoisii* by sequence alignment, and studied the conserved domain composition and evolutionary relationships among members of the MADS family.

In summary, based on PacBio Iso-Seq, this paper successfully obtained the basic data of the full-length transcriptome of *C. courtoisii*, and the MADS-box family genes of *C. courtoisii* were predicted and screened; thereby, providing a basic data reference for further studies on the biological functions of the MADS-box gene family, and for flower trait improvement, in Clematis varieties.

## Supporting information

**S1 Data.**
(ZIP)

## Author Contributions

**Data curation:** Linfang Li.

**Project administration:** Peng Wang.

**Resources:** Shu'an Wang.

**Writing – original draft:** Mingjian Chen.

**Writing – review & editing:** Mingjian Chen, Linfang Li, Ya Li.

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
