## [Decision Letter · Decision Letter 0]

14 Aug 2023

PONE-D-23-16625Screening and identification of MADS-box gene family in <clematis courtoisii=""> based on SMRT full-length transcriptome</clematis>PLOS ONE

Dear Dr. fang,

Thank you for submitting your manuscript to PLOS ONE. After careful consideration, we feel that it has merit but does not fully meet PLOS ONE’s publication criteria as it currently stands. Therefore, we invite you to submit a revised version of the manuscript that addresses the points raised during the review process.

We look forward to receiving your revised manuscript.

Kind regards,

Mojtaba Kordrostami, Ph.D.

Academic Editor

PLOS ONE

Journal Requirements:

"Funding This work was supported by the National Natural Science Foundation of China (31800603),  Jiangsu Key Laboratory for the Research and Utilization of Plant Resources (JSPKLB202203), and the Technology Innovation Alliance of Flower Industry Fund(2020hhlm004)."

"This work was supported by the National Natural Science Foundation of China (31800603), Jiangsu Key Laboratory for the Research and Utilization of Plant Resources (JSPKLB202203), and the Technology Innovation Alliance of Flower Industry Fund(2020hhlm004)."

"Funding This work was supported by the National Natural Science Foundation of China (31800603),  Jiangsu Key Laboratory for the Research and Utilization of Plant Resources (JSPKLB202203), and the Technology Innovation Alliance of Flower Industry Fund(2020hhlm004)."

7. Please amend the manuscript submission data (via Edit Submission) to include author "Mingjian Chen". 

8. Please amend your authorship list in your manuscript file to include author "Li Lin fang". 

**Additional Editor Comments:**

Dear Authors,

I hope this letter finds you well. Thank you for submitting your manuscript to PLOS ONE journal. Your work focusing on the genus Clematis and the MADS-box gene family's role in flowering regulation has been reviewed by two experts in the field. Based on their feedback and my evaluation, I have reached a decision regarding your manuscript.

Both reviewers recognize the potential value of your study, especially concerning the ornamental and breeding value of Clematis. The use of PacBio long-read sequencing to characterize total RNA from Clematis Courtoisii and define the transcriptomic landscape for the MADS-box gene family is noteworthy. However, they have raised several concerns that need to be addressed comprehensively before the manuscript can be considered for publication.

Key Feedback:

Reviewer 1:

Concerns about the clarity and depth of the introduction, particularly regarding the MADS-box gene family identification.

Questions about the replicates used in full-length transcriptome sequencing.

The method for identification and analysis of MADS-box genes was seen as not rigorous enough.

Clarity is needed in the methods and parameters for multi-sequence alignment and phylogenetic tree construction.

The figures provided are blurry and need to be clearer and more visually appealing.

The narrative lacks logical progression, and the relationship between transcriptome sequencing and gene family identification is not clear.

Reviewer suggests a more in-depth analysis, emphasizing the need for a deeper dive into the gene family identification.

Reviewer 2:

Concerns about the quality of figures and the necessity for high-quality uploads.

A lack of detail in the methodology, especially concerning de novo transcript assembly.

Questions about the significance of identifying lncRNAs.

Concerns about the functional annotation approach and its implications.

The need for a deeper interpretation and discussion regarding the phylogenetic analysis results.

In light of the above, I am returning your manuscript with a decision of "Revise". I believe that by addressing the reviewers' comments thoroughly, your manuscript can be enhanced significantly, making it a valuable contribution to our journal and the broader academic community.

Here's what we recommend for the revision:

Revisit the introduction, ensuring clarity and depth, particularly regarding the MADS-box gene family identification.

Provide clarity on the replicates used in full-length transcriptome sequencing.

Detail your methodology, especially concerning de novo transcript assembly and the identification and analysis of MADS-box genes.

Address the concerns raised about the figures and ensure they are of high quality and easily interpretable.

Revisit the narrative to ensure logical progression and clear relationships between different sections.

Dive deeper into the gene family identification, providing a comprehensive and in-depth analysis.

Address all other specific concerns raised by the reviewers.

Upon resubmission, please provide a detailed point-by-point response indicating how you have addressed each concern raised by the reviewers. This will greatly assist in the re-evaluation process.

We appreciate the time and effort you have put into your research and manuscript. I hope you find the reviewers' feedback constructive, and I look forward to receiving your revised manuscript.

Warm regards,

Mojtaba Kordrostami

Editor

PLOS ONE Journal

Reviewers' comments:

Reviewer's Responses to Questions

**Comments to the Author**

1. Is the manuscript technically sound, and do the data support the conclusions?

Reviewer #1: Partly

Reviewer #2: Partly

2. Has the statistical analysis been performed appropriately and rigorously? 

Reviewer #1: Yes

Reviewer #2: I Don't Know

3. Have the authors made all data underlying the findings in their manuscript fully available?

Reviewer #1: Yes

Reviewer #2: Yes

4. Is the manuscript presented in an intelligible fashion and written in standard English?

Reviewer #1: No

Reviewer #2: Yes

5. Review Comments to the Author

Reviewer #1: The genus Clematis (Ranunculaceae) is a vine with ornamental and breeding value, understanding the MADS-box gene family related to flowering regulation is of great significance to enhance the ornamental value of clematis. The author sequenced the full-length transcriptome of Clematis courtoisii and identified 19 members of the MADS-box gene family from the transcriptome. However, I don't think this manuscript can meet the requirements for publication, and it is suggested that the manuscript can be greatly revised.

L67-L70 The last paragraph of the introduction should include the MADS-box gene family identification.

L75 Just use a clematis? Are there no replicates set up in full-length transcriptome sequencing?

L114 The method of identification and analysis of MADS-box genes is not rigorous, usually in addition to using homologous genes for blatsp, it is necessary to identify the members of the gene family through the unique domain of the family, and check the integrity of the domain, and the family's conservative motif is not analyzed later, so I am skeptical of the results.

L123-L124 Please add the methods and parameters for multisequence alignment and phylogenetic tree construction.

L147 “CDS is a protein coding sequence that is completely consistent with the codon of the protein”, which is confusing.

L225 Please clarify the basis for the selection of the 97 MADS-box related genes.

L267 There does not appear to be a comparison of second- and third-generation transcriptomes in the manuscript.

All the pictures are blurry and can't see the legend clearly. The figures seem not attractive enough, especially figure 7 and 5, maybe the author should reorganize the complex information in a visually appealing way.

The writing lacks logic, and logical progressive relation of the content of full-length transcriptome sequencing and gene family identification is not clear, I think it is more appropriate to use transcriptome sequencing as the basis and premise of gene family identification, rather than describing the analysis and annotation of transcriptome data at length.

In general, the analysis of transcriptome sequencing results and gene family identification is relatively crude, lacks in-depth analysis, and is mostly descriptive results, although this manuscript named "Screening and identification of MADS-box gene family in based on SMRT full-length transcriptome", but the bioinformatics analysis of gene family identification is very simple, accounting for only a small part of the full text, and does not provide new information for understanding the evolution and function of MADS-box family (especially related to flowering regulation). Therefore, it is advised to add at least some common analysis for gene family identification (such as, the evolutionary relationship, chromosomal location, conserved structure, tissue expression pattern, as well as biological functions).

Reviewer #2: The authors have used PacBio long read sequencing to characterize total RNA from Clematis Courtoisii and define the transcriptomic landscape for the MADS-box gene family. Unfortunately a lot of detail is missing from the manuscript and I would like the authors to address my following concerns:

- The figures are of extremely poor quality and I am unable to read or interpret anything. Please upload high quality figures otherwise reviewing this manuscript properly will be impossible

- The authors mention that they do not use a genome or transcriptome reference but provide no detail on how they perform de novo transcript assembly. Detailed methods on how these transcripts were identified is important to assess the novelty of their results

- They identify lncRNAs as well as protein coding genes but do not provide reasons for why this may or may not be important

- The authors use hits against databases such as Pfam, KEGG etc. as a proxy for "functional annotation". They do not mention the background sets of the genes used which can severely affect the results obtained, and additionally do not provide any interpretation of function. This in my opinion is extremely misguiding to a reader.

- The authors perform phylogenetic analysis to identify MADS-box genes. Once again, they do not comment on the novelty or implications of their findings and their figure legends as well as methods are lacking with regard to interpretability.

Unless the authors address the major shortcomings outlined here, I am afraid I cannot recommend this manuscript as fit for publication.

6. PLOS authors have the option to publish the peer review history of their article (what does this mean?). If published, this will include your full peer review and any attached files.

Reviewer #1: No

Reviewer #2: No

---

## [Author Response · Author response to Decision Letter 0]

6 Oct 2023

Response to editor’s and reviewers’ Comments

Reviewer 1:

Concerns about the clarity and depth of the introduction, particularly regarding the MADS-box gene family identification.

Questions about the replicates used in full-length transcriptome sequencing.

The method for identification and analysis of MADS-box genes was seen as not rigorous enough.

Clarity is needed in the methods and parameters for multi-sequence alignment and phylogenetic tree construction.

The figures provided are blurry and need to be clearer and more visually appealing.

The narrative lacks logical progression, and the relationship between transcriptome sequencing and gene family identification is not clear.

Reviewer suggests a more in-depth analysis, emphasizing the need for a deeper dive into the gene family identification.

Reply: Thank you very much for your constructive comments. We have taken your kind suggestions. 

1. L67-L70 The last paragraph of the introduction should include the MADS-box gene family identification.

We have added the related introduction about MADS-box gene family identification in the L80-88 of the tracked version manuscript.

2. L75 Just use a clematis? Are there no replicates set up in full-length transcriptome sequencing?

The project was initiated with the objective of investigating the impact of three generations of sequencing in order to identify the genes associated with flower development in C. courtoisii. For sequencing, only C. courtoisii was chosen as the subject of research, and samples from different developmental stages were collected, ground, and mixed to serve as the material. We have added the related introduction in the L96-97 of the tracked version manuscript.

3. L114 The method of identification and analysis of MADS-box genes is not rigorous, usually in addition to using homologous genes for blatsp, it is necessary to identify the members of the gene family through the unique domain of the family, and check the integrity of the domain, and the family's conservative motif is not analyzed later, so I am skeptical of the results.

The genes related to the mads-box were initially screened based on the annotation results from the database in the L150-167 of the tracked version manuscript. Subsequently, these sequences were uploaded to the SMART online database for verification of their sequence structures, and further compared and validated using hmmer V3.0 software's hummscan function with MADS-box conserved domain pfm files PF00319 and PF01486. Ultimately, a total of 19 candidate MADS-box-associated genes were identified. Additional steps for analyzing conserved motif alignment with model plants are provided in the L180-187 of the tracked version manuscript.

4. L123-L124 Please add the methods and parameters for multisequence alignment and phylogenetic tree construction.

The method and parameter settings for multiple sequence alignment and phylogenetic tree construction using MEGA software are supplemented in the L169-178 of the tracked version manuscript.

5. L147 “CDS is a protein coding sequence that is completely consistent with the codon of the protein”, which is confusing.

We have revised the article by eliminating this section.

6. L225 Please clarify the basis for the selection of the 97 MADS-box related genes.

We modified the screening method and utilized the hmmscan function of the hmmer software to compare translated protein sequences of Unigenes against downloaded hmm files SRF-TF (PF00319) and K-box (PF01486), thereby identifying a total of nineteen directly associated MADS-box Unigenes. It should be noted that only MADS-box-related genes identified based on annotated information were used as reference.

7. L267 There does not appear to be a comparison of second- and third-generation transcriptomes in the manuscript.

The transcriptome sequencing of C. courtoisii was limited to a single third-generation analysis, lacking both a reference genome and a reference second-generation transcriptome. Consequently, the second-generation sequencing data from other clematis species were utilized for comparison and analysis.

8. All the pictures are blurry and can't see the legend clearly. The figures seem not attractive enough, especially figure 7 and 5, maybe the author should reorganize the complex information in a visually appealing way.

The requirements of the figure were carefully reviewed and necessary modifications were made to ensure the uploaded pictures met high quality standards.

 

Reviewer 2:

Concerns about the quality of figures and the necessity for high-quality uploads.

A lack of detail in the methodology, especially concerning de novo transcript assembly.

Questions about the significance of identifying lncRNAs.

Concerns about the functional annotation approach and its implications.

The need for a deeper interpretation and discussion regarding the phylogenetic analysis results.

Reply: Thank you very much for your constructive comments. We have taken your kind suggestions. 

1. The figures are of extremely poor quality and I am unable to read or interpret anything. Please upload high quality figures otherwise reviewing this manuscript properly will be impossible

The requirements of the figure were carefully reviewed and necessary modifications were made to ensure the uploaded pictures met high quality standards.

2. The authors mention that they do not use a genome or transcriptome reference but provide no detail on how they perform de novo transcript assembly. Detailed methods on how these transcripts were identified is important to assess the novelty of their results

We have added the parameters and Settings of the software used in the process to L102-131 of the tracking version of the manuscript.

3. They identify lncRNAs as well as protein coding genes but do not provide reasons for why this may or may not be important

We have revised the article by eliminating this section.

4. The authors use hits against databases such as Pfam, KEGG etc. as a proxy for "functional annotation". They do not mention the background sets of the genes used which can severely affect the results obtained, and additionally do not provide any interpretation of function. This in my opinion is extremely misguiding to a reader.

The quality of transcriptome sequencing data is crucial for accurate gene function annotation. High-quality data can enhance the reliability and precision of gene function annotation, thereby facilitating a better understanding of gene functions in organisms. However, the method used to screen C. courtoisii MADS-box related genes from gene functional annotations lacks rigor. In order to assess the transcriptome quality, we utilized the mentioned gene functional annotations (accounting for.19% of transcripts) as an indicator. To further search and identify MADS-box related genes, we modified our screening approach by employing hm V3.0 software's hummscan function to compare and validate conserved domain files PF00319 and PF01486 associated with MADS-box proteins. As a result, nineteen Unigenes directly linked to MADS-box were successfully identified. 

5. The authors perform phylogenetic analysis to identify MADS-box genes. Once again, they do not comment on the novelty or implications of their findings and their figure legends as well as methods are lacking with regard to interpretability.

we have made further modifications to the figure format in order to present a well-defined phylogenetic tree and facilitate comparison of conserved motifs. This will enhance our understanding of the type, quantity, and function of MADS-box genes associated with flower formation and development in C. courtoisii.

---

## [Decision Letter · Decision Letter 1]

31 Oct 2023

Transcriptome sequencing and screening of genes related to the MADS-box gene family in <clematis courtoisii="">

PONE-D-23-16625R1</clematis>

Dear Dr. Wang,

We’re pleased to inform you that your manuscript has been judged scientifically suitable for publication and will be formally accepted for publication once it meets all outstanding technical requirements.

Kind regards,

Mojtaba Kordrostami, Ph.D.

Academic Editor

PLOS ONE

Additional Editor Comments (optional):

The manuscript can be accepted now

Reviewers' comments:

Reviewer's Responses to Questions

**Comments to the Author**

1. If the authors have adequately addressed your comments raised in a previous round of review and you feel that this manuscript is now acceptable for publication, you may indicate that here to bypass the “Comments to the Author” section, enter your conflict of interest statement in the “Confidential to Editor” section, and submit your "Accept" recommendation.

Reviewer #1: All comments have been addressed

2. Is the manuscript technically sound, and do the data support the conclusions?

Reviewer #1: Yes

3. Has the statistical analysis been performed appropriately and rigorously? 

Reviewer #1: Yes

4. Have the authors made all data underlying the findings in their manuscript fully available?

Reviewer #1: Yes

5. Is the manuscript presented in an intelligible fashion and written in standard English?

Reviewer #1: Yes

6. Review Comments to the Author

Reviewer #1: My previous comments have been properly addressed.

My previous comments have been properly addressed.

7. PLOS authors have the option to publish the peer review history of their article (what does this mean?). If published, this will include your full peer review and any attached files.

Reviewer #1: No

---

## [Editor Report · Acceptance letter]

24 Jan 2024

PONE-D-23-16625R1 

PLOS ONE

Dear Dr. Li, 

I'm pleased to inform you that your manuscript has been deemed suitable for publication in PLOS ONE. Congratulations! Your manuscript is now being handed over to our production team.

Kind regards, 

on behalf of

Dr. Mojtaba Kordrostami 

Academic Editor

PLOS ONE